# The Relationship between Attachment, Dyadic Adjustment, and Sexuality: A Comparison between Infertile Men and Women

**DOI:** 10.3390/ijerph20043020

**Published:** 2023-02-09

**Authors:** Alessandra Santona, Laura Vismara, Laura Gorla, Giacomo Tognasso, Carolina Ambrosini, Anisa Luli, Luca Rollè

**Affiliations:** 1Department of Psychology, University of Milano-Bicocca, 20126 Milan, Italy; 2Department of Pedagogy, Psychology, Philosophy, University of Cagliari, 09123 Cagliari, Italy; 3Department of Developmental Psychology and Socialisation, University of Padua, 35131 Padova, Italy; 4Department of Psychology, University of Torino, 10124 Torino, Italy

**Keywords:** infertility, sexuality, attachment, dyadic adjustment, infertility couples

## Abstract

Infertility impacts several life dimensions. Among them, sexuality is particularly affected; yet studies have mainly focused on infertile women. We aimed to explore infertile men’s and women’s experiences in sexual satisfaction, internal control, and anxiety, and the relationship between attachment, dyadic adjustment, and sexuality. The sample consisted of 129 infertile people (47.3% females, 52.7% males, M_age_ = 39 years) who fulfilled an ad hoc questionnaire, the Multidimensional Sexuality Questionnaire (MSQ), the Experiences in Close Relationship-Revised (ECR-R), and the Dyadic Adjustment Scale (DAS). We found a significant effect of type of infertility and infertility factors on sexual anxiety only in infertile men. As regards infertile women, dyadic adjustment predicted sexual satisfaction, anxious attachment decreased sexual internal control, and avoidant attachment reduced sexual anxiety. As regards infertile men, high dyadic adjustment increased sexual satisfaction and a high avoidant attachment predicted high levels of sexual internal control. There was no relationship between attachment, dyadic adjustment, and sexual anxiety for infertile men. From the results, it emerges how important is to consider both dyadic adjustment and attachment in studying how infertility impacts women’s and men’s lives.

## 1. Introduction

### 1.1. Definition, Causes, and Treatment of Infertility

Infertility could be conceived as “a disease of the reproductive system defined by the failure to achieve a clinical pregnancy after 12 months or more of regular sexual intercourse (as there is no other reason, such as breastfeeding or postpartum amenorrhea). Primary infertility is infertility in a couple who has never had a child. Secondary infertility is failure to conceive following a previous pregnancy. Infertility may be caused by an infection in the man or woman, but often there is no obvious underlying cause” [1].

Infertility prevalence is increasing in both high- and low-income countries because of many socio-demographic reasons, such as living habits, nutritional factors, epidemic infections, and sexually transmitted pathologies and diseases [2]. National and international statistics indicate that infertility affects about 10–12% of couples worldwide, while in Italy, about 15% of couples suffer from this condition [3,4,5,6].

#### 1.1.1. Causes of Infertility

One crucial factor to consider in assessing a couple with fertility problems is the factor of age, mainly for the woman, whose fertility declines progressively from 30 years of age onward [6], while men are less affected. Organic or functional issues constitute 80% of other causes [6]. Additionally, poor lifestyle habits (i.e., use of tobacco, alcohol, drugs, anabolic steroids) are implicated as risk factors for infertility [5].

Around 40% of all infertile couples are usually characterized by a mix of factors (i.e., a female factor combined with a male disorder). Nevertheless, at least at the time of the diagnosis, approximately 15% of couples are classified as patients suffering from inexplicable or idiopathic infertility [7].

Part of idiopathic infertility may be explained by the role of mental disorders (i.e., stress, depression, addictions) that seem to modify the endocrine gland and immune system functioning at both the tissue and cellular level and are negatively linked to female and male fertility [8,9,10].

#### 1.1.2. Treatment of Infertility

In Italy, according to the European Society of Human Reproduction and Embryology (ESHRE), if a couple has an estimated live birth rate of 40% or higher per year, they are recommended to continue to seek pregnancy naturally [11], while under certain conditions, a course of stimulating medication, or Intrauterine Insemination (IUI) may be used. If these conservative medical treatments fail to achieve full-time pregnancy, the patients could undergo Assisted Reproductive Technology (ART) techniques, which include In Vitro Fertilization (called VF-ET ad In Vitro Fertilization and Embryo Transfer) and Intra Cytoplasmatic Sperm Injection (ICSI) [12].

### 1.2. Men’s and Women’s Experience of Infertility

Infertility has several psycho-sexual implications, as people with impaired fertility often undergo several stressful events and situations. Several factors contribute to the psychosocial outcome of infertility [13,14] as individuals diagnosed with infertility face an unsolvable condition with the available coping characterized by unpredictability, uncontrollability, and ambiguity [15].

Although the psychological, social, and sexual aspects of infertility and fertility treatment have been investigated comprehensively in women, the same factors relating to men are less well understood [16].

Stereotypically, women are presumed to desire children and therefore to experience grief when experiencing difficulties in having a baby, but men have been described as being “disappointed but not devastated” by the inability to have a child [17]. Contrary to these beliefs, some population-based investigations [18,19,20] found that there were no overall differences between women and men in the desire to have a child: indeed, men wanted children with similar intensity; thus, men do not experience infertility as merely “disappointing” but as a source of high stress.

Fisher and Hammarberg’s review [21] focused on infertile men’s experience and analyzed many dimensions of psychological functioning as the consequences of diagnosis on mood tone, identity, quality of life, and self-esteem. The results collected by the review are not homogeneous: overall, the prevalence of clinically significant psychological symptoms seems to be lower than the infertile men’s normative values [16]. Even if some studies indicated that infertility effects are more intensively experienced among women [22,23], some authors [24] concluded that is a result of stereotypes that state that men would manifest less adverse reactions to infertility than women. Indeed, even considering the probability of some gender-specific responses, both men and women have peculiar and important psychological needs as they are called to face an adverse life event like infertility.

### 1.3. The Repercussions of Infertility on the Couple’s Sex Life

Sexual health is directly associated with each individual’s psychological well-being and quality of life [23], as sexuality is inherent to human beings and intimately linked to their reproduction. Different dimensions of sexuality influence the relationships between partners: sexual satisfaction (which builds up the story of a couple’s relationship as a lover like expressing feelings to each other, frequent sexual activity, and desire together) [25], sexual anxiety or discomfort (regarding sexual behavior or performance may inhibit sexual interaction) [26] and the perception of control (this may play a role in attaining dyadic sexual satisfaction) [27].

Therefore, an interconnection between the unfulfilled desire for a child and sexual disorders or sexual discomfort can be observed in infertile couples [28,29]. A frequent example of discomfort in sexuality concerns the timing of intercourse: as the treatment of infertility imposes the couple to follow a precise sequence of sexual intercourse, the sexual moment loses its spontaneity but follows the “fertile window” as part of the treatment [30]. Hence, as the focus of sexual intercourse is on the act of procreation, behavior and the couple’s intimacy are altered [31]. Indeed, the sexual act is now conceived as a mechanical, scheduled, and timed process that is baby-oriented. This alteration often leads to lower sexual desire, lower satisfaction, and a lower perception of being able to control personal sexual life.

In addition, sexual disorders or other sexual dysfunction (SD) could rise in infertile couples [28,32]. These conditions can occur in both partners and might cause problems in every stage of sexual response [33], being a cause of infertility or triggered by it [34].

The most common SD is erectile dysfunction (ED) and ejaculatory disorders [28] in men, and genital/pelvic pain and penetration disorder in women [35]. These conditions can lead to frustration and unfulfilling sexual experiences for the partners [32] causing sexual performance anxiety that emphasizes them and therefore causes distress to the couple [32,36]. Ramezanzadeh and colleagues [37] found that despite a generally high level of sexual satisfaction in men who were attending an infertility clinic, 41.5% reported at least some reduction in sexual desire and 52.5% in satisfaction since they were diagnosed [37].

Conversely, according to a recent review [28], women are more likely to be affected by sexual disorders than men, and Marci and colleagues [38] revealed that the consequences of infertility on sexuality were worse in women than in men as women displayed lower scores in orgasm, sexual satisfaction, and the desire dimensions of sexuality. Nevertheless, even if women reported experiencing less sexual satisfaction compared to their partners and tended to avoid sexual intercourse more frequently [39], a recent interesting study [40] found that a significantly higher proportion of men (37%) than women (12%) described having received verbal pressure from their partner to engage in intercourse to conceive [40]. According to these results, sexual coercion during sexual intercourse for the purpose of procreation was associated for men, but not women, with distress on psychological and relational dimensions.

### 1.4. Adult Attachment and Sexuality

It is largely known that attachment concerns how each person reacts to stressful events [41,42] and its impact on sexuality is well recognized [43,44,45,46]; nevertheless, few studies have focused on the importance of attachment theory in order to explore infertile couples’ sexuality [45,47] or empirically examined the relationship between attachment and sexuality in the peculiar context of infertility [48].

In adulthood, attachment insecurity is conceptualized in attachment-related anxiety and avoidance, whilst secure attachment correlates with low levels of anxiety and avoidance [49]. In terms of attachment within the couple, attachment-related anxiety indicates a fear of abandonment by the partner and the consequent strong desire for proximity and reassurance. Indeed, while anxious people construct a negative perception of the self as unlovable, people having an attachment-related avoidance often experience discomfort with proximity and emotional intimacy with their partner, as well as excessive self-reliance and a negative model of others [49,50].

Examining the associations between attachment insecurity and sexuality [43,46], it was found that individuals high in anxiety tend to experience more negative emotions and worries about their attractiveness and sexual competence, becoming hypervigilant and easily disappointed during sex. On the other side, due to the discomfort, they experience in the context of intimacy, avoidant individuals tend to maintain an emotional distance and engage in sexual activities less frequently, and are dissatisfied with their sex life [51].

Focusing on fertility treatment, Purcell-Levesquea and colleagues [48] found that “women’s attachment-related avoidance, not anxiety, was related to greater sexual pain and lower sexual satisfaction” [48]. The same study indicated that in men higher anxiety levels were associated with more difficulties with erections and difficulties in reaching orgasm. This seems to prove that infertile people with insecure attachment would probably have trouble becoming involved with their body sensations during sex; women might alter their potentiality to lubricate, and men might have problems having an erection and consequently reaching orgasm.

Finally, significant correlations between men’s avoidance and their female partner’s difficulty to reach orgasm have emerged: in the context of fertility treatment, medically prescribed sexual intercourse combined with an avoidant male partner may negatively influence a woman’s sexual pleasure because it is not a priority for the couple [48,52].

### 1.5. Dyadic Adjustment and Sexuality

As infertility directly and strongly affects the sexual life and well-being of someone in a couple, it is crucial to evaluate how the couple itself reacts to infertility by understanding its dyadic adjustment.

Several studies that have explored the relationship between infertility and a couple’s adjustment reported different results: some suggest that infertile women experienced more problems in terms of marital harmony than infertile men [34,53,54], while others reported the opposite [55]. Leety and colleagues [56] noted that, if infertility was due to a single member of the couple, women reported more distress in marital harmony, whereas, if infertility was due to both members or it was not traceable to any cause, marital harmony didn’t differ between men and women.

Also, in relation to sexuality, no significant association between sexuality and marital trouble in either fertile or infertile couples, was found. Nevertheless, both infertile men and women had lower “consensus” and “affective expression” subscale scores on the Dyadic Adjustment Scale (DAS) than male and female subjects of the control group. Furthermore, compared to women in the same group, infertile men reported higher DAS satisfaction scores while they didn’t differ in terms of the other DAS scores [57,58].

Tuzer and colleagues [59] found decreased sexual desire in women: in particular, men showed a significantly higher affectional expression than women who displayed increased levels of trait anxiety in the same domain (i.e., sexual desire and expressions of love).

Another recent study that used the Dyadic Adjustment Scale to explore the dynamics in couples receiving infertility treatment, found that women reported lower marital adjustment and quality of life than men: indeed, except for the DAS’s subscale of sexual satisfaction, the males mean scores in this scale were higher than those of females [60].

To the best of our knowledge, there aren’t specific studies focused on the peculiar influence of dyadic adjustment on sexuality in infertile couples. Through our research, we aim to explain the possible effect of infertile couples’ dyadic adjustment on their quality of sex life.

### 1.6. Aims of the Study

The literature has highlighted that infertility is a life crisis that entails several challenges for those who suffer from this condition [8,15,61]. One specific dimension directly affected by infertility is sexuality as discovering they are infertile changes how a person experiences her/his sexuality both individually and in the couple [8,28,62]. For this reason, it is crucial to examine the complex connection between infertility and sexuality. Moreover, as the literature has been focused especially on women’s experience of infertility and its consequences on sexuality, we believe that is necessary to consider infertile men’s sexual experiences as well.

Finally, as sexuality is also influenced by dimensions such as attachment and dyadic adjustment [51,55,60], we also considered these two aspects in the current research.

The present study intends to explore the experiences of infertility in the sexual lives of men and women by focusing on three specific aspects of sexuality particularly relevant in the context of infertility and its treatment: sexual anxiety, internal control, and satisfaction. In addition, we aim to understand the connection between attachment and dyadic adjustment of the couple and their influences on sexuality, in both infertile women and infertile men.

The novel aspect of the current study is the exploration of three specific aspects of sexuality and their connection with attachment and dyadic adjustment in an infertile sample.

According to the studies previously conducted on the current theme, we expected that:(a)Some specific aspects of infertility (i.e., factor and type of infertility and type of treatment) decrease sexual satisfaction and internal control while increasing sexual anxiety in both infertile women and infertile men.(b)High levels of dyadic adjustment predict higher sexual satisfaction and internal control and lower levels of sexual anxiety in the two groups.(c)High levels of anxiety and avoidance in attachment raise sexual anxiety and reduce sexual satisfaction and internal control.

## 2. Materials and Methods

### 2.1. Participants and Procedure

A total of 129 Italian infertile people, 47.3% (N = 61) females and 52.7% (N = 68) males, aged between 26 and 57 years (M = 39.13; DS = 6.7; M_females_ = 37.4 (DS = 6.4); M_males_ = 40.6 (DS = 6.6)) participated in our study. Our sample was not composed of couples. More detailed sociodemographic information is reported in Table 1.

The sample was recruited both in hospitals and in public and private centers for Medically Assisted Reproduction (MAR) in Northern Italy. We contacted physicians and psychologists working in these centers to explain the research and agree on timings and data collection methods. After that, physicians and psychologists contacted infertile couples and administered the questionnaires during the medical check-ups following the infertility diagnosis and through the treatment of infertility.

Data were collected in 2017, before COVID-19, so fertility treatments were not altered by the pandemic situation.

Data collection was carried out following the provisions of Italian law 196/2003 in collecting the participants’ consent, and all the questionnaires were anonymous. Before beginning the questionnaire, participants received both an oral and a written explanation of the study from a research assistant and a Doctor or Physician. The research was previously approved by the Ethics Committee of the Psychology Department of Milano-Bicocca University (protocol code 0029119/13, 16/10/2013) and was handled according to the Declaration of Helsinki.

### 2.2. Measures

Participants completed the following instruments:

An ad hoc questionnaire was created to collect participants’ socio-demographic information and aspects regarding infertility. In particular, it investigated:-type and factor of infertility-center consulted for the treatment-type of treatment-who decided to start the treatment-people informed about the treatment decision-thoughts about adoption

The Dyadic Adjustment Scale (DAS) [63] is a 32-item self-report instrument that measures dyadic adjustment and measures each partner’s representation of the relationship by exploring four dimensions. DAS is composed of four subscales: Dyadic Consensus (13 items capturing agreements or disagreements between the two partners regarding different topics); Dyadic Cohesion (5 items, measuring how often partners share pleasant time and activities); Affectional Expression (4 items, showing how couples express and communicate feelings, love, and sexuality) and Dyadic Satisfaction (10 items, providing a measure of overall satisfaction and happiness for the relationship). Moreover, the instrument also gives a total score of dyadic adjustment with a range from 0 to 151 with higher scores indicating more positive dyadic adjustment. Typically, cut-off scores between 92 to 107 are used to discern between distressed and non-distressed couples.

We obtained a Cronbach’s alpha of 0.823 for the total score of DAS, 0.803 for Dyadic Consensus, 0.768 for Dyadic Cohesion, 0.552 for Affectional Expression, and 0.362 for Dyadic Satisfaction. While the Cronbach alphas of the total score, dyadic consensus, and dyadic cohesion were adequate, we discovered only moderate reliability of the two alphas of Affectional Expression and Dyadic Satisfaction that should be considered.

The Multidimensional Sexuality Questionnaire (MSQ) [64] is a 60-item self-report scale that measures psychological dimensions linked to individual sexual life. Specifically, items are rated on a 5-point Likert scale and participants are asked to point out how much the item represents their sexual characteristics.

The instrument is constituted by 12 subscales including sexual motivation (e.g., “I am very motivated to be sexually active”), preoccupation (e.g., “I think about sex all the time”), assertiveness (e.g., “I am very assertive about the sexual aspects of my life”), depression (e.g., “I am disappointed about the quality of my sex life”), anxiety (e.g., “I feel anxious when I think about the sexual aspect of my life”), self-esteem (e.g., “I am a pretty good sexual partner”), monitoring (e.g., “I sometimes wonder what others think of the sexual aspects of my life”), internal control (e.g., “My sexuality is something that I am largely responsible for”), external control (e.g., “Most things that affect the sexual aspects of my life happen to me by accident”), consciousness (e.g., “I am very aware of my sexual feelings”), satisfaction (e.g., “I am very satisfied with the way my sexual needs are currently being met”), and fear (e.g., “I sometimes have a fear of sexual relationships”).

Out of the 12 dimensions obtainable by the MSQ, we focused our attention only on 3 subscales: sexual anxiety, sexual internal control, and sexual satisfaction.

We obtained Cronbach’s alphas on these dimensions of 0.700, 0.677, and 0.818, respectively.

The Experiences in Close Relationship-Revised (ECR-R) [65,66] is a 36-item self-report instrument that measures feelings and behaviors linked to attachment in romantic relationships. Participants are asked to fulfill the items using a 7-point Likert scale (from 1 “strongly disagree” to 7 “strongly agree”), with higher scores revealing higher endorsement of the construct. Through the instrument, romantic attachment can be classified in two dimensions: Avoidance of intimacy (level of preoccupation related to sharing emotional closeness, i.e., “I prefer to not show my partner how I feel deep down”), and Anxiety about abandonment (measures the preoccupation with the relationship or the need for intimacy, i.e., “I worry about being alone”). In our sample, we obtained a Cronbach’s alpha of 0.475 for the Avoidance dimension and 0.450 for the Anxiety, finding only moderate reliability that should be considered.

### 2.3. Analysis Plan

Firstly, we conducted descriptive statistics and Pearson bivariate correlations, and Fisher’s Z tests among all the research variables considering both infertile men and women. We then performed an independent *t*-test to compare the two groups.

Secondly, we ran linear regressions to explore the relationship between infertility aspects, dyadic adjustment, attachment, and sexuality. We differentiated all the analyses according to gender, dividing infertile women from infertile men. We used the statistical software IBM SPSS version 28 (IBM Corp, Armonk, NY, USA) for all the analyses.

## 3. Results

### 3.1. Descriptive Analysis and Bivariate Correlations

Table 2 reports all the information about infertility we collected in both the infertile women and infertile men groups. It can be seen that a consistent segment of the participants suffered from an inexplicable type of infertility and do not know the specific factor of their fertility problems. Another interesting aspect is that most of the participants revealed not having considered adoption as a possible option for becoming a parent.

Table 3 reports the bivariate correlations between the variables we considered for the study and the Fisher’s Z test. In infertile women, ECR anxiety was significantly related to sexual internal control, while ECR avoidance significantly correlated to sexual anxiety. We also found a significant correlation between dyadic adjustment and sexual satisfaction. In infertile men, ECR anxiety did not correlate with any of the variables considered, while avoidance was positively connected to sexual internal control. The dyadic adjustment correlated with sexual satisfaction also for the infertile men’s group. Fisher’s Z tests indicated that infertile women and men did not significantly differ in these correlations, except for the correlation between anxiety in attachment and internal sexual control (Z = −2.207, *p* = 0.014) and sexual satisfaction (Z = −1.77, *p* = 0.038).

We then performed an independent *t*-test to compare the scores of infertile women and men with respect to the variables considered, and the two groups did not differ regarding their attachment, dyadic adjustment and sexual anxiety, internal control, and satisfaction (Table 4).

### 3.2. Multiple Regressions

A multiple linear regression has been performed to verify the hypothesis that aspects of infertility would affect sexual satisfaction, internal control, and anxiety in both groups.

#### 3.2.1. Infertility and Sexual Life

Our results showed that type of infertility, infertility factor, type of treatment, and thoughts about adoption did not affect infertile women’s sexual life in any of the aspects we considered. Indeed, no relation between infertility aspects and sexual ones was statistically significant. Conversely, we found a significant effect (F (4,40) = 2.879, *p* = 0.035) of type of infertility and infertility factors on sexual anxiety in infertile men (see Table 5).

#### 3.2.2. Dyadic Adjustment, Attachment, and Sexual Life

As we wanted to understand the relationship between dyadic adjustment, personal attachment, and sexuality, we performed linear regressions in the two groups (Table 6). As regards infertile women, our results showed that higher levels of dyadic adjustment predict higher levels of sexual satisfaction (B = 0.148, *p* < 0.001). Moreover, the presence of anxious attachment decreases sexual internal control (B = −0.116, *p* = 0.033) and a high avoidant attachment reduces infertile women’s sexual anxiety (B = −0.094, *p* = 0.012).

As regards infertile men, we found that high levels of dyadic adjustment increase sexual satisfaction (B = 0.092, *p* = 0.036) and high avoidant attachment predicts high levels of sexual internal control (B = 0.116, *p* = 0.033). Our results showed no relationship between attachment, dyadic adjustment, and sexual anxiety for infertile men.

## 4. Discussion

Infertility is a life crisis that involves several challenges [8,15,61]; a specific dimension of life that seems to be strongly impacted by infertility is sexuality as discovering they are infertile changes how a person experiences sexuality both individually and within the couple [8,28,62]. Nevertheless, the results of the various studies in the literature regarding sexuality in infertile couples and the difference between women and men are contradictory.

Wischmann and colleagues [67] found that 500 couples starting infertility therapy reported no difference in satisfaction with their sex lives over and against the norm, even if the men reported slightly higher sexual discomfort than women [67]. Conversely, in a study conducted on 144 couples in the process of beginning in vitro fertilization (IVF) treatment, Slade and colleagues reported that the women explicit significantly higher dissatisfaction with their sex lives than their male partners, although within the clinical norm [68]. Moreover, in his survey, Möller [69] reports that 50% of his sample (considerably more women than men) claimed that they modified their sex lives as a result of the unfulfilled desire for a child. In particular, two-thirds indicate that their sex lives have declined, and one-third experience at least an initial intensification [69]. Finally, other studies report that couples wishing for a child claim to experience sexual pleasure and frequent sexual intercourse to a larger extent than the corresponding norms [70].

Although the two groups did not differ regarding their sexual dimensions (anxiety, internal control, satisfaction) and personal aspects, such as attachment and dyadic adjustment, multiple regressions show significant differences with respect to gender. Indeed, women’s factors associated with infertility such as the type of infertility, the type of treatment, and thoughts about adoption did not affect their sexual life in any of the aspects considered in our study, while a significant effect of type of infertility and infertility is present for sexual anxiety in infertile men. These results confirm the results found by Nachtigall and colleagues [71] who showed that men with male factor infertility experienced more negative emotional responses, including a sense of loss, stigma, and reduced self-esteem than men whose partners were infertile or who were in couples suffering from unexplained infertility. Furthermore, the men in the infertile couples had higher levels of depressive symptoms and anxiety than did fertile men [72]. These findings could be explained by several reasons. If infertility lasts for a long time, sexual relationships could become more closely associated with experiences of failure. This unpleasant feeling of worthlessness and lack may also impact patients’ perception of their bodies and their reproductive function, though this was observed to be more likely among women than men [73]. Additionally, during infertility treatment, some couples perceive the medical team as symbolically present during sexual intercourse, and the pressure of time and the purpose of “baby-making” make sexual intercourse very difficult, usually because of erectile dysfunction. This initial erectile dysfunction, aside from increasing the sexual anxiety in infertile men, may turn into a persistent sexual disorder because of a “vicious circle” effect which manifests in the following steps: “performance anxiety—inhibition—erectile dysfunction—the feeling of shame and failure—performance anxiety”.

While studies about the impact of infertility on sexuality are numerous, research about the impact of dyadic adjustment and attachment on sexuality in infertility couples is scarce [24,47,55]. As we aimed to understand the connection between attachment and dyadic adjustment of the couple and their influences on sexuality, we believe that this constitutes the novelty of the current study.

In both women and men of our sample, the results showed that higher levels of dyadic adjustment anticipated higher levels of sexual satisfaction. Even if we didn’t find studies that focused on the specific influence of dyadic adjustment on sexuality in infertile couples, Güleç and colleagues [55] showed that, within an infertile group, the men scored higher DAS satisfaction than women; however, there wasn’t a difference between infertile men and women in terms of the other DAS dimension scores [55]. Tashbulatova [57,74] reported that, in general, couple’s adjustment positively influenced sexual functioning in couples, but it may be even more true for infertile couples, who showed higher levels of marital harmony than fertile ones: in order to safeguard marriage, these partners have to cope for long periods with the crisis and treatment of infertility, thinking together about decisions to be made and sharing support and affection. A high marital harmony and a good dyadic adjustment between the partners who go through the condition of infertility could strengthen their union also in the sexual sphere leading to an increase in sexual satisfaction [61].

Regarding the relation between ECR attachment and the dimensions of sexuality, we found slightly different results between the two groups.

First, the presence of anxious attachment decreases sexual internal control in women. If partners have been trying to have a baby for a long time, individuals with high levels of anxiety can be burdened by feelings of defeat, performance concerns [48], and apprehension of losing their partner. Because of their difficulty in emotional regulation [48,75], for anxious women, it might be very hard to connect with their body sensations during sexual intercourse. Additionally, as we know that the moments and number of intercourses are often planned by the treatment, women with an anxious attachment may lose the feeling of sexual internal control as their sexuality could be perceived as something that no longer depends on their choice.

Conversely, it is largely known that individuals with high levels of avoidance generally strive to maintain emotional distance [48]; this strategy could explain the lower level of sexual anxiety in women and the higher levels of sexual internal control in men. Avoidance is surely linked with the need to be in control of situations [48] and this may well be generalized to sexuality.

One interesting result of our study is that we did not discover a significant relationship between attachment, dyadic adjustment, and sexual anxiety for infertile men. Moreover, our results showed a non-significant relationship also between attachment anxiety and sexual anxiety for both infertile women and infertile men. These results are in line with the literature reporting that avoidant attachment could be more relevant in influencing infertile people’s sexual life rather than anxiety [48]. One possible explanation could be linked to the psychological constitutional elements of the anxious attachment itself. Attachment-related anxiety has been defined as a fear of abandonment by the partner, which leads the anxious person to request (and desire) physical proximity and reassurance from the other partner. When facing a fertility issue, the physical proximity could be preserved (or also increased with the aim to have a baby) and, for this reason, people with anxious attachment could not experience an increase of sexual anxiety as, conversely, people with avoidant attachment could.

Moreover, adult romantic relationships are characterized by three motivational systems (attachment, sexuality, and caregiving) that could be strongly influenced by individual characteristics, meaning that some specific connection between them could not be fully detected.

Nevertheless, more in-depth studies about these aspects in infertile men and women are needed, as it would be recommended to evaluate the connection between sexuality and attachment by combining standardized instruments with attachment in-depth interviews.

Certainly, there are some limitations to this study. First, it has a cross-sectional design, which doesn’t allow us to conclude a causal relationship. Second, we did not consider all the dimensions implicated in sexuality, but we decided to focus only on those dimensions (sexual anxiety, satisfaction, internal control) that offer a global idea of individual sexuality; thus, we may miss further relevant dimensions within infertile couples undergoing treatment.

Third, we did not consider other essential aspects in analyzing the impact of infertility on sexuality: in fact, we did not focus on social support or the support within the couple dealing with childlessness.

Another limit that should be taken into consideration is the sample size: indeed, the relatively small number of participants suggests the need to conduct more studies on the comparison between infertile men and women and to plan some follow-up studies for this sample. However, as far as we know, the current study aimed at examining aspects not yet explored, so its results should be considered a valuable starting point for future research on this topic.

Finally, longitudinal studies applying statistical methods for paired data are needed to fully understand the impact of infertility on sexual life focusing on its development and changes over time.

## 5. Conclusions

To the best of our knowledge, whereas sexual problems in infertile people have been largely investigated [8,28,62], the influence of dyadic adjustment and romantic attachment on sexuality is still not well known. This is even more true for men; indeed, the current study contributes to filling the gap in the literature with respect to men who experience infertility.

From the results of our study, a good dyadic adjustment emerges as important to maintain the sexual relationship satisfactorily. Furthermore, it seems that anxious attachment in infertile women impacts negatively their sexual internal control, while avoidant attachment leads to lower sexual anxiety in women and higher sexual internal control in men.

The knowledge gained and the concepts explained in this study could help both researchers and clinicians in their dealings with people who are facing infertility. These results can also be useful to guide the development of psychological interventions as they highlighted the close connection between the personal, couple, and sexual dimensions for people who are dealing with a diagnosis of infertility.

## Figures and Tables

**Table 1 ijerph-20-03020-t001:** Sociodemographic information.

**Marital Status**	
Married	72.1% (N = 93)
Unmarried	27.9% (N = 36)
**Education**	
Other (primary school or Ph.D. degree)	5.4% (N = 7)
Master’s degree	18.6% (N = 24)
Bachelor’s degree	13.2% (N = 17)
High school degree	41.9% (N = 54)
Middle school degree	20.9% (N = 27)
**Professional activity**	
Working in executive professions in an office	30.2% (N = 39)
Intellectual and scientific professions	22.5% (N = 29)
Technical professions	18.6% (N = 24)
No qualified professions	28.7% (N = 37)

**Table 2 ijerph-20-03020-t002:** Infertility information: type of infertility, infertility factors, ongoing treatments, and adoption thoughts.

** *Type of Infertility* **		
Primary	Women	50.8% (31)
	Men	54.4% (37)
Secondary	Women	4.9% (3)
	Men	4.4% (3)
Inexplicable	Woman	44.3% (27)
	Men	41.2% (28)
** *Infertility Factor* **		
Male factor	Women	11.5% (7)
	Men	16.2% (11)
Female factor	Women	18% (11)
	Men	22.1% (15)
Couple factor	Women	14.8% (9)
	Men	14.7% (10)
Unknown factor	Women	55.7% (34)
	Men	47% (32)
** *Type of Treatment Ongoing* **		
ICSI	Women	26.2% (16)
	Men	25% (17)
VF-ET	Women	24.6% (15)
	Men	22.1% (15)
Intrauterine insemination	Women	6.6% (4)
	Men	7.4% (5)
Meropur assumption	Women	3.3% (2)
	Men	4.4% (3)
Assisted conception	Women	1.6% (1)
	Men	5.9% (4)
No treatment	Women	37.7% (23)
	Men	35.3% (24)
** *Thoughts about Adoption* **		
Yes, only if treatments will be ineffective	Women	34.4% (21)
	Men	30.9% (21)
Yes, in any case	Women	8.2% (5)
	Men	1.5% (1)
Never	Women	57.4% (35)
	Men	67.6% (46)

**Table 3 ijerph-20-03020-t003:** Correlations and Fisher Z Test Coefficients for sexual dimensions, attachment, and dyadic adjustment in infertile women and men.

	Sexual Anxiety	95% CI	Internal Sexual Control	95% CI	Sexual Satisfaction	95% CI
**Women (N = 61)**						
Anxiety	0.069	[−0.187, 0.315]	−0.319 *	[−0.529, −0.073]	−0.144	[−0.382, 0.112]
Avoidance	**−0.320 ***	[−0.533, 0.079]	0.185	[−0.070, 0.417]	0.190	[0.065, 0.422]
Dyadic Adjustment	−0.235	[0.460, 0.018]	0.044	[−0.210, 0.293]	**0.425 ****	[0.194, 0.611]
**Men (N = 68)**						
Anxiety	0.067	[−0.179, 0.306]	0.068	[−0.179, 0.306]	0.173	[−0.074, 0.400]
Avoidance	−0.166	[−0.394, 0.081]	**0.247 ***	[0.003, 0.463]	0.205	[0.041, 0.428]
Dyadic Adjustment	−0.227	[−0.444, 0.016]	0.125	[−0.120, 0.357]	**0.309 ***	[0.072, 0.512]
**Fisher’s Z test**						
	**Z**	** *p* **	**Z**	** *p* **	**Z**	** *p* **
Anxiety	0.011	0.496	−2.207	**0.014**	−1.77	**0.038**
Avoidance	−0.908	0.182	−0.36	0.359	−0.086	0.466
Dyadic Adjustment	−0.047	0.481	−0.452	0.326	0.744	0.229

* *p* < 0.01; ** *p* < 0.005.

**Table 4 ijerph-20-03020-t004:** Table of comparisons between infertile women and men for sexual, attachment, and adjustment dimensions.

Variable	Gender		t	d	*p*
	Women (N = 61)	Men (N = 68)			
	Mean (SD)	Mean (SD)			
Anxiety	61.26 (9.79)	58.32 (8.59)	1.80	0.32	0.074
Avoidance	86.85 (9.41)	86.59 (10.75)	0.145	0.02	0.885
Dyadic Adjustment	112.73 (12.11)	111.05 (12.32)	0.775	0.13	0.440
Sexual Anxiety	7.02 (2.67)	7.82 (3.25)	−1.52	−0.26	0.131
Internal Sexual Control	14.46 (3.94)	14.73 (4.27)	−0.374	−0.06	0.709
Sexual Satisfaction	19.61 (4.32)	18.39 (4.41)	1.57	0.27	0.118

**Table 5 ijerph-20-03020-t005:** Linear regressions with sexual dimensions as dependent variables and infertility aspects as independent variables.

Sexual Satisfaction	Women					Men				
	β	SE	95% CI	t	*p*	β	SE	95% CI	t	*p*
Type of infertility	0.028	0.858	[−1.611, 1.861]	0.145	0.885	−0.134	0.984	[−2.699, 1.277]	−0.723	0.474
Infertility factor	−0.252	0.700	[−2.319, 0.513]	−1.291	0.205	0.123	0.724	[−0.997, 1.929]	0.643	0.524
Treatment	0.068	0.275	[−0.438, 0.675]	0.430	0.669	0.256	0.293	[−0.104, 1.079]	1.66	0.104
Adoption thoughts	−0.046	0.678	[−1.564, 1.182]	−0.281	0.780	0.078	0.638	[−0.964, 1.615]	0.510	0.613
**Sexual Internal Control**										
Type of infertility	0.099	0.768	[−1.156, 1.954]	0.519	0.607	−0.052	0.947	[−2.167, 1.662]	−0.266	0.791
Infertility factor	−0.146	0.627	[−1.739, 0.798]	−0.751	0.458	0.127	0.697	[−0.966, 1.852]	0.635	0.529
Treatment	−0.049	0.246	[−0.576, 0.421]	−0.314	0.755	−0.072	0.282	[−0.695, 0.445]	−0.444	0.659
Adoption thoughts	−0.222	0.607	[−2.062, 0.397]	−1.370	0.179	0.011	0.615	[−1.201, 1.283]	0.066	0.948
**Sexual Anxiety**										
Type of infertility	0.166	0.438	[−0.499, 1.273]	0.884	0.382	0.521	0.683	[0.682, 3.443]	3.020	**0.004**
Infertility factor	0.168	0.357	[−0.411, 1.035]	0.873	0.388	−0.385	0.503	[−2.109, −0.078]	−2.176	**0.036**
Treatment	−0.087	0.140	[−0.362, 0.206]	−0.557	0.580	−0.143	0.203	[−0.615, 0.207]	−1.002	0.322
Adoption thoughts	−0.073	0.346	[−0.857, 0.544]	−0.452	0.654	0.159	0.443	[−0.399, 1.392]	1.120	0.269

**Table 6 ijerph-20-03020-t006:** Linear regressions with sexual dimensions as dependent variables and attachment and dyadic adjustment as independent variables.

Sexual Satisfaction	Women (N = 61)					Men (N = 68)				
	β	SE	95% CI	t	*p*	β	SE	95% CI	t	*p*
Anxiety	−0.074	0.055	[−0.143, 0.077]	−0.601	0.551	0.249	0.065	[−0.001, 0.258]	1.991	0.051
Avoidance	0.150	0.057	[−0.045, 0.183]	−0.601	0.230	0.241	0.053	[−0.006, 0.205]	1.891	0.063
Dyadic Adjustment	0.415	0.042	[0.064, 0.232]	3.532	**<0.001**	0.256	0.043	[0.006, 0.178]	2.146	**0.036**
**Sexual Internal Control**										
Anxiety	−0.289	0.053	[−0.223, −0.010]	−2.187	**0.033**	0.166	0.065	[−0.047, 0.214]	1.272	0.208
Avoidance	0.092	0.055	[−0.072, 0.149]	0.695	0.490	0.291	0.053	[0.010, 0.223]	2.185	**0.033**
Dyadic Adjustment	0.026	0.041	[−0.073, 0.090]	0.206	0.837	−0.066	0.043	[−0.064, 0.110]	0.527	0.600
**Sexual Anxiety**										
Anxiety	−0.049	0.035	[−0.084, 0.057]	−0.381	0.705	0.033	0.050	[−0.088, 0.113]	0.251	0.803
Avoidance	−0.331	0.036	[−0.167, −0.021]	−2.580	**0.012**	−0.115	0.041	[−0.117, 0.047]	−0.853	0.397
Dyadic Adjustment	−0.225	0.027	[−0.103, 0.004]	−1.843	0.071	−0.196	0.033	[−0.119, 0.015]	−1.547	0.127

## Data Availability

The data presented in this study are available upon request from the corresponding author.

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
