# Peer review of "The Relationship between Attachment, Dyadic Adjustment, and Sexuality: A Comparison between Infertile Men and Women"

_ijerph, 2023, doi:10.3390/ijerph20043020_

Round 1

Reviewer 1 Report (Previous Reviewer 1)

The revision has improved the quality of the paper by a great deal.

Author Response

Thank you very much for all your previous comments, we are happy that the revision has improved the quality of our paper.

I attached here a Word with the changes we made.

This manuscript is a resubmission of an earlier submission. The following is a list of the peer review reports and author responses from that submission.

Round 1

Reviewer 1 Report

- The Introduction is too long.  The subsections on causes of infertility (1.1.1) and tretament of infertility (1.1.2) seem unnecessary in an article on the impact of subfertile couples on their sexuality.  Indeed, subfertility is a better term than infertility: only men with azoospermia and women with certain conditions such as proven menopause or uterine agenesis are infertile.  Fertility is now seen in terms of per-year or per-lifetime chance of a couple to achieve conception.

In the Introduction, the authors use sexual terminology that is now considered obsolete and even painful.  Vaginismus is now considered to be part of the "genital/pelvic pain and penetration disorder" (DSM-5). 

Some sentences are nonsensical: "in men higher anxiety levels were associated to more erections and orgasm's difficulties" (159-160).  In the next sentence we learn that pain alters "their potential to lubricate (women), have an erection (men), and come to orgasm (men)" (162-163).  Apparently, women only need to lubricate, the men must come.  It may be that these end-points were the results of the study, but it remains an unpleasant sentence.

A bit later the same nonsense: "women" (with subfertility?) "possessed" (sic) "lower marital adjustment and quality of life than men: ... the males mean scores were higher than those of females" (189-191).  It might work for an article about rabbits, less for humans with subfertility.

Some sentences, in the Italian way, lack a personal noun: "Is largely known" (137) or "wasn't observed" (183). 

- In the Materials and Methods, it is unclear why the authors sometimes opted for the total score (Dyadic adjustment) and sometimes for a few items (in the MSQ "we focused our attention only on three subscales" (284-285)).  Why?  Again, in the ECR-R three (of 36?) items were distilled: avoidance and anxiety.  This results in two anxiety scales: ECR anxiety and MSQ sexual anxiety (table 4).  For the non-expert, it all remains a bit muddy.

In table 1, we get some very detailed description of educational levels and professional activities ("workers of fix and mobile machinery and vehicle drivers"), which I am sure can be shortened into three or four groups. 

- In the Results, I fail to understand the difference between "type of infertility: inexplicable" (in my view, there is only primary and secondary, if you have kids or not) and "infertility factor: unknown"; or what "couple factor" means (compounded male and female factor?).

In Table 3, we are presented with correlations of items we have not yet met before.  It is unclear why there is no correlation between "anxiety" and "sexual anxiety".

In table 4, the five items are presented in random order as "table of comparisons", apparently between men and women but not matched for couple status.

Tables 5 and 6 present "linear regressions": a lot of figures with uncertain meaning.

It remains unexplained why 61 women and 68 men were included, and why the focus was not on intra-couple comparisons.  The rather small amount of participants is an additional limitation, in my view (433-443).     

Reviewer 2 Report

Thank you for the opportunity to review the article entitled: "The relationship between attachment, dyadic adjustment, and sexuality: a comparison between infertile men and women". The manuscript tells about an important topic and is well-written. The introduction is a very valuable part of the article. The authors presented there in an accessible way the current knowledge on psychological factors related to infertility. They did it in a way that allows people who had so far little known about it to become acquainted with the topic. As a result, the article can be read and understood also by people who are not familiar with its subject so far. The authors also described some of the research tools they used very well, thanks to which the article can be a source of information for other researchers interested in measuring the same variables. However, the article has several aspects that could be improved to make it even better. Below I present my recommendations for changes and corrections.

1. Authors often introduce paragraphs of only one sentence into the text. In many cases, these sentences could be made part of the next paragraph. For example, a paragraph on lines 38-39 (one sentence) and a paragraph on lines 40-42 (also one sentence) could be combined with the next paragraph. Similar changes can be made in many places in the text.

2. In section 1.1.1. the authors included three paragraphs. The first and last of these briefly describe the various factors associated with infertility. However, the second paragraph only signals the involvement of mental disorders and immune system factors without giving examples of specific relationships. I recommend adding 1-2 sentences explaining the relationship between mental disorders, immune system factors, and infertility.

3. In section 1.1.2. the authors introduce some professional terms and their abbreviations. However, some abbreviations do not have the full names indicated (IFV, FIVET, ICSI). I recommend adding the full names of these technologies so that a reader who does not know the meaning of these abbreviations can understand them.

4. In Table 1, the authors present information on the education and professional activity of the respondents. However, there is no information in the text indicating that these data are of any importance from the point of view of the conducted research. I encourage the authors to explain why they presented this information and whether it is important from the point of view of the presented research. Otherwise, consider deleting this information as redundant to the research and article topic.

5. The authors indicated several limitations of their study. However, they did not pay attention to the fact that one of the important limitations is a small research sample of only 61 women and 68 men. For many readers, the presentation of the results of the study collected in such a small sample may be unconvincing. Moreover, the authors themselves in the Discussion refer to the results of the study by Wischmann et al. (2001) including 500 pairs, and Slade et al. (1997) covering 144 couples. In this context, the results presented by the authors look relatively less credible.

6. In Table 3, the authors show the correlations between the variables in the groups of men and women. Some of the correlations are significant in one group but not in the other group. However, the authors did not show the results of the Fisher z test for whether the differences in correlations between men and women are significant. I recommend conducting the Fisher z test and presenting its results. With such a small sample, it may turn out that the differences in correlation coefficients between men and women are insignificant.

7. Concerning the multiple regression analyses performed, I have two remarks. First, the authors present the B and beta coefficients but do not present standard errors and 95% confidence intervals. I think that in such a small research sample showing 95% C.I. is very important because it should be shown that 95% of C.I. does not contain the value 0.00. The presence of 0.00 in 95% C.I. even with significant results, would indicate that the results should be treated with caution. Second, I suggest the authors consider performing the calculations in a slightly different way, which would allow them to avoid some of the problems arising from the fact that they use two small samples (61 and 68 participants) in their analysis. If there were no significant differences in t-tests between men and women (Table 4) and the results of Fisher's tests showed non-significant differences in correlation coefficients (Table 3), perhaps instead of separate regression analyses for men and women, a multiple regression analysis could be performed in the combined sample of 129 people, where gender would be one of the predictors (0 = men; 1 = women). Another idea is to analyze moderation, where gender would be the moderator of the relationship between the predictors and the dependent variables. Perhaps the methods of analysis suggested above will allow presenting the collected results more conclusively. I suggest the authors consider such ways of analyzing the relationships between the variables.

8. Finally, I would like to mention an issue that was unclear to me in the context of regression analyses in Table 5. The authors list such predictors as the type of infertility, infertility factor, treatment, and adoption thought. The possible values of these variables are shown in Table 2. In my opinion, the adoption thought can be easily converted to a linear numerical value (e.g. 0 = never, 1 = Yes, only if treatments will be ineffective; 2 = Yes, in any case). However, the remaining responses regarding infertility factors and treatment are difficult to linearize (e.g. what value should be given to "unknown factor" for infertility factor?). So I would like the authors to describe exactly in the manuscript what numerical values they gave to each answer option in case of questions from their "Ad hoc questionnaire". To be able to linearize a qualitative variable in regression analysis, this variable must be either bivalent (e.g. man/woman) or there must be a clear direction, where lower and higher values can be easily indicated (e.g. in the case of level of education). The authors should show that such linearization is possible concerning the variables they measured in the "Ad hoc questionnaire".

I think that the article is interesting, presents valuable results, and is worth to be published. However, I believe that it should be changed in line with the above-mentioned recommendations. As some of the recommended changes involve performing additional calculations, my recommendation is: "Reconsider after major revision".